# Assessments of effectiveness of technologies utilizations in VIHSCM among selected health facilities in Tanzania mainland

**Henry A. Mollel[1]\*, Lawrencia D. Mushi[1], Richard V. Nkwera[2]**

**1** Department of Health Information Systems, Mzumbe University, Morogoro, Tanzania, **2** Centre of Excellence in Health Monitoring and Evaluation, Mzumbe University, Morogoro, Tanzania

\* hemollel@mzumbe.ac.tz

## Abstract

Immunization coverage remains a challenge in many developing countries Tanzania being no exception. The current increase in technology adoption in the immunisation supply chain promises the attainment of universal health coverage and Sustainable Development Goals (SDGs) on immunisation. This study evaluates the effectiveness of technology integration in Vaccine and Immunization Health Supply Chain Management (VIHSCM) in Tanzania. This study adopted an exploratory descriptive cross-sectional design. The study collected data using structured questionnaires from health facilities that adopted VIHSCM technologies in Arusha, Mwanza, Morogoro and Mbeya regions, Tanzania. Data were analysed using descriptive statistics and cross-tabulations with the aid of the Statistical Package of Social Sciences 23rd Edition (SPSS). The study included 37 health facilities in Tanzania, mainly district hospitals (59.5%). Respondents were mostly female (70.3%), averaging 45 years old, with 1–5 years of immunization experience. While all facilities had refrigerators, digital reporting tool usage was low, with many relying on paper forms. District hospitals and health centres had higher digital tool adoption rates compared to dispensaries. Despite the under-utilization of systems like ILS, TImR, and GoTHOMIS, digital tools were deemed crucial for vaccine supply management. While District Hospitals report high relevance of digital tools, Health Centres and Dispensaries show moderate relevance. Challenges include incomplete technology adoption, inadequate infrastructure, and variable perceptions of technology effectiveness. Digital technologies significantly improve vaccine and immunization supply chain management, particularly in larger facilities. Technologies like the Tanzania Immunization Registry (TImR) and Integrated Logistics Systems (ILS) enhance data accuracy and efficiency. Addressing facility-specific challenges and increasing investment in digital tools are crucial for optimizing vaccine supply chains and achieving immunization targets in Tanzania. Future research should involve larger samples to generalize findings and further explore technology impacts on VIHSCM.

**Data Availability Statement:** Data are available for open access via the Inter-university Consortium for Political and Social Research (ICPSR), openICPSR,

at the following link: https://doi.org/10.3886/E209303V1.

**Funding:** The authors declare that this work was funded by the University of Rwanda, East African Regional Centre of Excellence for Vaccines, Immunisation, and Health Supply Chain Management (UR EAC RCE-VIHSCM), Research Grant Number RCE-VIHSCM 002/2021, through the Research Grant Scheme to All Authors (HM; RM & RM). However, the funder had no direct involvement in the study design, data collection and analysis or writing of this manuscript.

**Competing interests:** The authors have declared that no competing interests exist.

## 1. Introduction

Immunization Supply Chain and Logistics is crucial for the Immunization Agenda 2030 (IA2030), which outlines a global strategy for vaccines and immunisation for 2021–2030 [1]. IA2030 emphasises integrating immunisation into primary health care to achieve universal health coverage and the health-related Sustainable Development Goals (SDGs) [2]. Despite the historical success of routine immunization programs, national vaccine supply chains continue to encounter persistent challenges due to the introduction of new vaccines, the need for adaptation to evolving delivery strategies, and the growing demand for advanced cold chain technologies [3, 4]. Consequently, sustained investment and continuous innovation in the Immunization Supply Chain and Logistics (ISCL) are essential to sustaining and amplifying the impact of vaccination programs [5, 6]. National immunisation programs are transforming supply chains by integrating new technologies and innovative methods for Effective Vaccine Management (EVM) to keep pace with the changing landscape of immunisation programmes [4, 7–9]. The WHO-UNICEF assessment highlights the crucial role of modern technologies in strengthening vaccine supply chains [2, 10]. Effective technology utilization, including advanced inventory tracking and cold chain management, is vital for optimizing vaccine distribution and improving coverage, especially in low- and middle-income countries [11–13].

Despite being recognized as one of Africa's best-performing immunisation programs, Tanzania prioritises improving its national immunization efforts [12, 14]. The Ministry of Health and Social Welfare (MoHSW) developed the National Immunization Strategy (NIS) for 2021–2025, aligning with the Health Sector Strategic Plan V, Immunisation Agenda 2030, and Gavi 5.0 [15]. The NIS aims to deliver lifelong protection through high-quality, equitable immunization services and to integrate a resilient program into primary healthcare [14, 15]. To support this, the MoHSW's Immunization and Vaccines Development (IVD) department implemented advanced technologies, including the Vaccine Information Management System (VIMS), Warehouse Management Information Systems, and the Tanzania Immunization Registry (TImR), to enhance supply chain and logistics [11, 12, 16].

However, despite these efforts, some regions in Tanzania still report immunisation coverage below 80%, reflecting disparities across the country [17]. Tanzania DHS reported a decline in full vaccination rates for basic antigens from 73% in 1991–2016 to 53% in 2022, while the percentage of unvaccinated children aged 12–23 months fluctuated between 2% and 5% [18]. The decline in fully vaccinated children since 2015–16 is mainly due to limited capacity, inaccurate target populations, poor vaccine supply visibility, and challenges in tracking immunisation defaulters worsened by the COVID-19 outbreak, [4, 7, 8, 18]. The Tanzania National Immunization Strategy (2021–2025) identified critical challenges, such as inadequate monitoring devices, incomplete temperature charts, and insufficient responses to alarms (MOH, 2020; UNICEF, 2020). Similarly, Both the 2015 Effective Vaccine Management Assessment and a recent study of 57 GAVI-eligible countries reveal a global decline in vaccine handling, with less than 25% meeting maintenance and stock standards and only 29% ensuring proper temperature control [4, 19]. Additionally, challenges such as inadequate electricity, unreliable network connectivity, limited information on cold chain equipment availability, and a lack of trained health personnel have been acknowledged as obstacles in integrating technologies for vaccine management in health facilities [7, 9, 20–22]. Addressing these challenges and evaluating the efficacy of current technologies in vaccine supply chains remains a critical area of inquiry, and this study significantly contributes to the existing literature by assessing the effectiveness of technology utilization in Vaccine and Immunization Health Supply Chain Management (VIHSCM) at primary facilities in Tanzania.

## 2. Design and method

### 2.1. Study setting

A facility-based exploratory descriptive cross-sectional design approach was employed to understand the effectiveness of utilization of technologies for VIHSCM across health facilities in three purposively selected regions of mainland Tanzania namely Arusha, Mwanza, Morogoro and Mbeya. This research design allowed both qualitative and quantitative data to be collected simultaneously using a structured questionnaire, analysed, and later combined for interpretation to provide a comprehensive analysis of the effectiveness of the utilization of technologies for VIHSCM across health facilities.

### 2.2. Data collection instruments and sampling procedures

Data collection was conducted in three purposively selected regions of mainland Tanzania namely Arusha, Mwanza, Morogoro and Mbeya. In each of the selected regions, two districts were purposively selected. The criteria for selection of both regions and districts were the urban-rural divide and the rate of vaccines and immunization uptake rate. The selection was based on capturing contextual variations that can influence the uptake, scale and integration of adopted technologies and innovations into existing systems and policies.

In the selected regions and councils, two and three relevant officials were selected purposively based on involvement in the implementation of vaccines and immunization activities. These included the Regional Medical Officer, Regional Immunization and Vaccine Officer, District Medical Officer, District Immunization and Vaccine Officer, and District HMIS (MTUHA) Focal Person. In each of the selected health facilities, three staff were purposively selected based on their involvement in vaccine and immunization management. These were the health facility in-charge, the in-charge of Reproductive and Child Health (RCH), and the RCH vaccine coordinator.

### 2.3. Data collection

Primary data were collected using a Structured questionnaire and Secondary data were collected through documentary review. A structured Questionnaire was administered to 37 respondents at the health facility level.

### 2.4. Data processing and analysis

Data collected during the study was cleaned, processed and analysed using IBM Statistical Package for Social Studies (SPSS). Because of the small number of records, the analysis only focused on descriptive statistics (frequencies, percentages, means, and standard deviation). It was performed for all demographic and health facilities characteristics as well as for study variables including health worker's understanding of the presence and utilization of technologies for VISHSCM.

### 2.5. Ethical considerations

Ethical clearance was obtained from the National Institute for Medical Research. The research was also approved by Mzumbe University. Permission to conduct research in the relevant institutions was obtained from the Ministry of Health (MOH) and the President's Office of Regional Administration and Local Government (PORALG). Participants were given a consent form describing the purpose of the study and their position to participate or terminate their participation even during the interview. Only consented participants were interviewed.

## 3. Results

### 3.1. Health facility and respondents' characteristics

The findings show that 37 respondents from enrolled health facilities gave complete responses making a response rate of 100 percent. The study assessed health facilities in four regions Arusha, Morogoro, Mbeya, and Mwanza regions. A total of 37 respondents from health facilities, in 8 districts of 4 regions participated in this study. The majority of participants were from district hospitals, accounting for 59.5% of the study population. Health centres and dispensaries accounted for smaller proportions, with 10.8% and 29.7%, respectively. The study shows that 11 (29.7%) respondents were male while 26 (70.3%) of the respondent were female. The age distribution of the respondents across health facilities showed that the average age was 45 years old. About 29.7 were RCH in charge coordinators with facility RCH in charge 24.3%, medical officer in charge, MTUHA, Medical Facility In charge, Facility In charge nurse, and Facility In charge. The results show that 17 (45.9%) had been working on Vaccines and Immunization activities for about 5 years, 6–10 years of Experience (27%), while (16.2%) had more than 10 years of experience and (10.8%) had less than year Experience in Vaccines and Immunization Management. (Table 1).

**Table 1. Demographic and Institutional characteristics across the health facility level.**

| Variable | Frequency(n) | Percentage(%) |
|---|---|---|
| **Sex** | | |
| Male | 11 | 29.7 |
| Female | 26 | 70.3 |
| Total | 37 | 100 |
| **Age** | | |
| 25–34 | 12 | 32.4 |
| 35–44 | 12 | 32.4 |
| 45–54 | 10 | 27 |
| 55–64 | 3 | 8.1 |
| Total | 37 | 100 |
| **Facility Level** | | |
| District Hospital | 22 | 59.5 |
| Health Center | 4 | 10.8 |
| Dispensary | 11 | 29.7 |
| **Job Title** | | |
| RCH in charge Coordinator | 11 | 29.7 |
| Medical officer In charge | 8 | 21.6 |
| MTUHA | 5 | 13.5 |
| Facility incharge | 2 | 5.4 |
| Facility RCH In charge | 9 | 24.3 |
| Medical Facility Incharge | 1 | 2.7 |
| Facility In charge Nurse | 1 | 2.7 |
| **Duration(Years)** | | |
| < 1 | 4 | 10.8 |
| 1–5 | 17 | 45.9 |
| 6–10 | 10 | 27 |
| >11 | 6 | 16.2 |
| **Total** | **37** | **100** |

**Table 2. Technology used for vaccines and immunization storage.**

| Technologies | District Hospital | Health Centre | Dispensary | Total |
|---|---|---|---|---|
| Refrigerator | 22(59.5%) | 4(10.8%) | 11(29.7%) | 37 (100%) |

The data on duration of employment indicates that the largest proportion of individuals (45.9%) have been employed for 1–5 years, suggesting a turnover rate that may be common in healthcare settings. Approximately 27.0% have been employed for 6–10 years, indicating a notable presence of individuals with moderate tenure. A smaller proportion of individuals (16.2%) have been employed for over 11 years, indicating longevity in their positions. Only 10.8% of individuals have been employed for less than 1 year, suggesting a relatively smaller proportion of newcomers to their roles. (Table 1)

### 3.2. Status of technology utilizations for VIHSCM across health facilities

The Data shows that in all 37 visited health facilities six visited district councils reported the presence of refrigerators (Table 2).

About 13.6% and 25% of respondents reported the predominantly use of software for reporting with District Hospitals and Health Centres respectively indicating a relatively low adoption rate compared to other methods at the District Hospital (Table 3). About (25%) to (72.7%) of respondents in health centres and Dispensaries respectively reported that the majority of reporting in all facility types is done using forms. while No software was used for reporting in Dispensaries. A significant portion of facilities, particularly District Hospitals, employ both software and forms for reporting reflecting efforts to leverage the benefits of both digital and paper-based reporting systems to ensure comprehensive data capture (Table 3).

### 3.3. Tools for vaccines and immunization management

About (9.1%) of respondents reported that District Hospitals employ electronic software for vaccine administration. While (25%) of respondents reported the usage of electronic software in Health Centres, indicating a greater uptake of digital tools at this level. About (18.2%) of respondents reported that similar to District Hospitals, Dispensaries also exhibit a lower adoption rate of electronic software for vaccine administration (Table 4).

A majority (45.5%) of respondents reported that District Hospitals rely on traditional paper-based forms for administering vaccines. A similar trend is observed, with (50%) of respondents reporting that Health Centres utilise paper forms as the primary tool for vaccine administration. While (54.4%) of respondents reported that Dispensaries exhibit the highest reliance on paper-based forms for vaccine administration among the facility types (Table 4).

In managing supplies and logistics of vaccines and immunization services at the health facility about (25%) of the respondents reported that Integrated Logistics Systems (ILS) are Utilized by Health Centres, about (4.5%) and (27.3%) of the respondents reported that the Tanzania Immunization Registry (TImR) is primarily used in District Hospitals and

**Table 3. Tools used to report the used vaccines.**

| | District Hospital | Health Centre | Dispensary | Total |
|---|---|---|---|---|
| Software | 3 (13.6%) | 1 (25%) | 0 | 4 (10.8%) |
| Form | 9 (40.9%) | 1 (25%) | 8(72.7%) | 18 (48.6%) |
| Both software and form | 10 (45.5%) | 2 (50%) | 3 (27.3%) | 15 (40.5%) |
| Total | 22 (100%) | 4(100%) | 11(100%) | 37(100%) |

**Table 4. Tool used to administer vaccine (B10).**

| Technology | District Hospital | Health Centre | Dispensary | Total |
|---|---|---|---|---|
| Electronic software | 2(9.1%) | 1(25%) | 2(18.2) | 5(13.5) |
| Form | 10(45.5%) | 2(50%) | 6(54.4%) | 18(48.6%) |
| Both electronic software and form | 10(45.5%) | 1(25%) | 3(27.3%) | 14(37.8%) |
| | | | | 37(100) |

Dispensaries respectively. About (4.5%) and (18.2%) of respondents reported DHIS 2 is used in District Hospitals and Dispensaries, respectively. About (18.2%) and (9.1%) of reported GoTHOMIS are used in District Hospitals and Dispensaries. The Data show that respondents reported a significant proportion of facilities, especially Health Centres (75%) and Dispensaries (45.5%), do not use any software for logistics management (Table 5).

### 3.4. Relevancy and effectiveness of technology utilizations in VIHSC management

The Data show that a significant majority of respondents (62.2%) consider the tools and software to be very relevant in facilitating the timely supply of vaccines in District Hospitals. While a notable portion of respondents (37.8%) mainly in Health centres and Dispensary perceives the tools as moderately relevant (Table 6).

The Data in Table show that about (18.2%) and (27.3%) of respondents reported technology has contributed to effective management in a significant portion mainly districts, Hospitals, and Dispensaries respectively. A considerable number of respondents (36.4%), (50%), and (36.4%) reported increased access to vaccines as a result of technological integration in District Hospitals, Health Centres and Dispensaries. While (50%) of respondents reported the highest increase in vaccine availability in Health Centres. Only a small number of respondents (9.1%) reported improved monitoring of vaccine ordering in Dispensaries. About (9.1%) of respondents have reported benefits in maintaining vaccine quality in District Hospitals. About (18.2%) of respondents reported benefits in monitoring vaccine temperature in Dispensaries. About (9.1%) of Respondents reported that Technology has facilitated Direct Communication Between Facility and DIVO in the District hospitals (Table 7).

### 3.5. Tendencies of vaccines stock out and ordering across health facilities levels

The Data show that about (40.9%) of District Hospitals experience stock-outs for specified vaccines once a year, while the majority (59.1%) never experience stock-outs. About Half of the Health Centres (50%) report experiencing stock-outs once a year and the other half (50%) never experience stock-outs. The majority of Dispensaries (63.6%) report experiencing stock-

**Table 5. VIHSCM tool for supplies & logistics management (B10).**

| Technology | District Hospital | Health Centre | Dispensary | Total |
|---|---|---|---|---|
| Integrated Logistics Systems (ILS) | 0 | 1 (25%) | 0 | 1 (2.7%) |
| Tanzania Immunization Registry (TImR) | 10(45.5%) | 0 | 3(27.3%) | 13 (35.1%) |
| DHIS 2 | 1(4.5%) | 0 | 2(18.2%) | 3(8.1%) |
| GoTHOMIS | 4(18.2) | 0 | 1(9.1%) | 5(13.5%) |
| No software | 7(7) | 3(75%) | 5(45.5%) | 15(40.5%) |
| | | | | 37(100) |

**Table 6. Relevancy of the tool or software in facilitating timely supply of vaccines (13).**

|  | District Hospital | Health Centre | Dispensary | Total |
|---|---|---|---|---|
| Very Relevant | 16 (72.0%) | 2 (50%) | 5 (45.5%) | 23(62.2%) |
| Moderate relevant | 6 (27.3%) | 2 (50%) | 6 (54.5%) | 14(37.8) |
| Total | 22 (100%) | 4 (100%) | 11(100%) | 37(100%) |

outs for specified vaccines once a year, while a smaller proportion (36.4%) never experience stock-outs. The Data also show that Across all facility types, the duration from ordering to delivery of vaccines takes only a few days (Table 8).

## 4. Discussion

This study highlighted the effectiveness of technology in children's vaccine and immunization supply chain management. The study findings indicate that health workers are employing technologies for the storage and reporting of vaccines in health facilities. Literature supports that electronic systems optimize the reporting process, conserve time and resources, and enhance the tracking of vaccine inventories, scheduling, and monitoring, thereby improving overall vaccine management [21, 23].

The findings indicate that technology integration in health facilities is incomplete. Specifically, the Integrated Logistic System (ILS) is not employed in dispensaries and health centres. The Tanzania Immunization Registry (TImR), while operational in some areas, is only partially implemented across 3,736 facilities in 15 regions. TImR is functional in Arusha and Mwanza but is not utilized in Mbeya, with its implementation halted in Meru and Longido due to equipment malfunctions [7]. The Government of Tanzania Health Management Information System (GoTHOMIS) is exclusively employed at the District Council hospital level, with limited application in Health Centres or Dispensaries. This is consistent with findings from studies conducted in Tanzania, South Africa, and India, which reveal that health facilities transitioning from paper-based systems to digital platforms for vaccine ordering, administration, and reporting encounter significant challenges. These challenges include inadequate internet connectivity, frequent power outages, and inconsistencies between reported vaccine stocks and physical inventory counts [7, 9, 21, 23]. These issues highlight the need for clear national policies to guide the digital transition. Despite these challenges, the Tanzanian government remains committed to fully implementing digital health systems by the end of 2021 to improve efficiency, data quality, and system reliability in immunization services.

The findings indicate that digital tools and software are broadly recognized as crucial for the timely delivery of vaccines, particularly in larger facilities like District Hospitals (Table 6).

**Table 7. The way technology has assisted in increasing access and utilization of vaccines and immunization (B6).**

|  | District Hospital | Health Centre | Dispensary | Total |
|---|---|---|---|---|
| Effective management | 4(18.2%) | 0 | 3(27.3) | 7 (18.9) |
| Increased access of vaccines at the facility | 8(36.4%) | 2(50%) | 4(36.4%) | 14 (37.8%) |
| Increased availability of vaccines at the facility | 6(27.3%) | 2(50%) | 1(9.1%) | 9(24.3) |
| Increased monitoring of vaccine ordering | N/R | 0 | 1(9.1%) | 1(2.7%) |
| Maintain the quality of the vaccine | 2(9.1%) | 0 | N/R | 2(5.4%) |
| Monitoring of temperature | N/R | 0 | 2 (18.2%) | 2(5.4%) |
| Provide direct communication between the facility and DIVO | 2 (9.1%) | 0 | 0 | 2(5.4%) |
| Total |  |  |  | 37(100%) |

**Table 8. Tendencies of vaccines stock out and ordering across health facilities levels.**

| | | District Hospital | Health Centre | Dispensary | Total |
|---|---|---|---|---|---|
| **How often do you get out of stock for the specified vaccines?** | Once a year | 9(40.9) | 2 (50) | 7(63.6) | 18(48.6) |
| | Never experienced stock-out | 13(59.1) | 2 (50) | 4(36.4) | 19(51.4) |
| **How long does it take from ordering to delivery?** | Takes only few days | 22 (100) | 4(100) | 11(100) | 37(100) |

Conversely, some Health Centres and Dispensaries view these technologies as having only moderate relevance, suggesting that there are opportunities for enhancing their effectiveness and integration [7, 24]. The success of digital tools in vaccine supply chains shows how important they are for strengthening immunization efforts. This highlights the need for ongoing investment and improvements in these systems. Additionally, addressing the specific challenges of different healthcare facilities is crucial for improving vaccine logistics and immunization outcomes in Tanzania, as noted by [7, 23, 25] and [24]

The findings presented in Table 6 indicate that technology substantially enhances vaccine access and utilization by improving management practices, increasing availability, ensuring quality, and enabling better communication. However, the adoption and impact of specific technological interventions differ among facility types. District Hospitals and Health Centres experience more significant benefits in terms of improved access and availability, whereas Dispensaries gain advantages in effective management and temperature monitoring. Research suggests that the digital transformation of immunization logistics has markedly strengthened immunization programs by streamlining health workers' tasks, enhancing data quality and utilization, minimizing stock-outs, and boosting coverage. By 2021, electronic immunization registries (EIR) had been piloted or implemented in over 50 low- and middle-income countries [4, 7, 9, 24].

The findings highlight that digital tools and software are largely viewed positively for ensuring a reliable vaccine supply across healthcare facilities in Tanzania. District Hospitals generally find these tools effective, while Health Centres and Dispensaries show more varied opinions. Addressing these differences and making continuous improvements to the tools are crucial for optimizing vaccine supply chain management and improving immunization outcomes. The findings in Table 7 reveal that, although most healthcare facilities encounter occasional vaccine stock-outs, the rapid delivery turnaround times demonstrate effective supply chain management. This underscores the critical need to address stock-out issues and ensure timely delivery to sustain high immunization coverage and meet public health objectives. This observation aligns with previous research, which highlights the importance of these factors in achieving effective immunization programs [7, 8, 20, 21, 23].

## 5. Conclusion

This study highlights the effectiveness of digital technologies in improving children's vaccine and immunization supply chain management in Tanzania. The transition from paper-based to digital systems has significantly enhanced data accuracy, efficiency, and vaccine management, despite ongoing challenges such as poor internet connectivity, battery issues, and mismatched vaccine stocks. Technologies like the Tanzania Immunization Registry (TImR), Integrated Logistic System (ILS) and the Government of Tanzania Health Management Information System (GoTHOMIS) have been implemented in various facilities, though nationwide adoption is still incomplete.

VIHSCM have proven highly relevant in larger facilities like District Hospitals, though their relevance in Health Centres and Dispensaries indicates areas for improvement.

Addressing these challenges can further optimise vaccine supply chains and improve immunization outcomes. The government's commitment to nationwide digital health implementation aims to enhance efficiency, improve data quality, reduce workload, and ensure continuous system usage. Overall, the study underscores the need for continuous investment in and enhancement of digital tools to improve vaccine supply chain management and immunization outcomes in Tanzania. Addressing the specific challenges faced by different facility types will further optimize the integration of immunization programs into primary healthcare systems, thereby contributing to the achievement of universal health coverage (UHC) and the fulfilment of health-related Sustainable Development Goals (SDGs).

## 6. Research limitations

Data collections were conducted in three regions and six councils; therefore, our findings, conclusion and recommendations reflect the settings and technologies applied in those councils. Using a small sample size might make it difficult to draw conclusions and generalize. However, the findings can be replicated in other contexts since the context in which Vaccination and Immunization of VIHSCM processes do not vary significantly and are guided by similar operational frameworks. Despite these limitations, the study's direct observation study design can show an actual practice. vaccine and cold chain management practices. Future research needs to be conducted with a larger sample of health facilities and health workers in the country to be able to realise and generalise the effectiveness of technology utilisation technologies for VIHSCM.

## Acknowledgments

The authors thank the top leadership of Mzumbe University for their technical support and for facilitating collaboration with the University of Rwanda and the implementation of this study. We are also thankful to Noel Otieno, Irene Moshi and Rehema Mgoda for their support as research assistance. Last but not least, we thank everyone who participated and contributed to this study.

## Author Contributions

**Conceptualization:** Henry A. Mollel.

**Data curation:** Lawrencia D. Mushi, Richard V. Nkwera.

**Formal analysis:** Richard V. Nkwera.

**Investigation:** Henry A. Mollel.

**Methodology:** Henry A. Mollel, Richard V. Nkwera.

**Project administration:** Henry A. Mollel.

**Resources:** Lawrencia D. Mushi.

**Software:** Richard V. Nkwera.

**Supervision:** Henry A. Mollel.

**Validation:** Henry A. Mollel.

**Writing – original draft:** Henry A. Mollel, Lawrencia D. Mushi, Richard V. Nkwera.

**Writing – review & editing:** Henry A. Mollel.

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
