## [Decision Letter · Decision Letter 0]

2 Apr 2024

PGPH-D-23-02110

Assessments of Effectiveness of Technologies Utilizations in VIHSCM Among Selected Health Facilities in Tanzania Mainland

Dear Dr. Mollel,

Thank you for submitting your manuscript to PLOS Global Public Health. After careful consideration, we feel that it has merit but does not fully meet PLOS Global Public Health’s publication criteria as it currently stands. Therefore, we invite you to submit a revised version of the manuscript that addresses the points raised during the review process.

In your revisions please address the reviewers comments as well as the following:

1. Please make sure you address the relevance of the study for immunization programs beyond Tanzania, by clearly identifying the gaps in literature and the ways this article is filling those gaps. 

2. Please provide more background on the Tanzanian context in general, and more information on the supply chain management system in particular.

3. The article would benefit from more focus on the policy implications of the study in Tanzania and beyond.  

We look forward to receiving your revised manuscript.

Kind regards,

Anat Rosenthal

Academic Editor

Journal Requirements:

1. We ask that a manuscript source file is provided at Revision. Please upload your manuscript file as a .doc, .docx, .rtf or .tex.

Additional Editor Comments (if provided):

Reviewers' comments:

Reviewer's Responses to Questions

**Comments to the Author**

1. Does this manuscript meet PLOS Global Public Health’s publication criteria? Is the manuscript technically sound, and do the data support the conclusions? The manuscript must describe methodologically and ethically rigorous research with conclusions that are appropriately drawn based on the data presented.

Reviewer #1: Yes

Reviewer #2: Yes

2. Has the statistical analysis been performed appropriately and rigorously?

Reviewer #1: I don't know

Reviewer #2: No

3. Have the authors made all data underlying the findings in their manuscript fully available (please refer to the Data Availability Statement at the start of the manuscript PDF file)?

Reviewer #1: Yes

Reviewer #2: No

4. Is the manuscript presented in an intelligible fashion and written in standard English?

Reviewer #1: Yes

Reviewer #2: No

5. Review Comments to the Author

Reviewer #1: This paper deals with an important topic with applied implications. I have indicated that this manuscript as it stands, needs a major revision. The reasons for my recommendation are two-fold: 1. There are multiple stylistic and grammatical errors and inconsistencies which undermine the findings being documented. I have highlighted in yellow some of these grammatical inconsistencies in the attached file. As a first step, someone needs to go through the entire manuscript and do a thorough check of the writing style and grammar. 2. The authors have presented a textual representation of the numbers that are already presented in tables; in other words, this is duplication of information, which readers might find unnecessary. I strongly suggest that rather repeating the numbers in a paragraph that are already in a table, the authors should focus more on analyzing the data and the implications of their findings. The findings of this study are interesting, but the conclusions are rather bland in that the authors need to say a lot more about what these data mean for Tanzania's policy and logistics concerned with making its vaccination program highly effective. I would be happy to review a revised version of this manuscript.

Reviewer #2: This is a helpful study relevant to policymakers in Tanzania, and I’m so glad that health workers found the supply chain management system useful.

As the manuscript is written, it does not really articulate what this research is adding to the literature or knowledge on this topic that would be useful or relevant to researchers working on immunization more broadly. This could help elevate this paper to broaden its interest to those beyond Tanzania.

The manuscript could benefit from a more targeted articulation of what information this adds to the literature and what the broad take-aways are. Some suggestions to this end:

• The introduction would be strengthened by clearly articulating the gap in academic knowledge that this article is filling. The practical utility of this information is clear—but what is this paper adding to the academic literature? What are the gaps in literature in supply chain management broadly, beyond Tanzania (and perhaps even beyond immunization)?

• The manuscript would benefit from some information on the supply chain management system itself; since it seems to be useful, readers from other countries will want that detail.

• I was interested in some more detailed analysis that could help policymakers in Tanzania and elsewhere. Did satisfaction with the system differ, for example, depending on whether the system was electronic or on paper?

• The tables are somewhat confusing as laid out; I was not always sure what the percentages were referring to. For example in Table 2 I initially thought that not all facilities had refrigerators; I think reformatting the tables could help with this.

• Qualitative interviews were mentioned but not included in the manuscript. What insights come from that material?

6. PLOS authors have the option to publish the peer review history of their article (what does this mean?). If published, this will include your full peer review and any attached files.

**Do you want your identity to be public for this peer review?** For information about this choice, including consent withdrawal, please see our Privacy Policy.

Reviewer #1: No

Reviewer #2: No

---

## [Decision Letter · Decision Letter 1]

22 Jul 2024

PGPH-D-23-02110R1

Assessments of Effectiveness of Technologies Utilizations in VIHSCM Among Selected Health Facilities in Tanzania Mainland

Dear Dr. Mollel,

Thank you for submitting your manuscript to PLOS Global Public Health. After careful consideration, we feel that it has merit but does not fully meet PLOS Global Public Health’s publication criteria as it currently stands. Therefore, we invite you to submit a revised version of the manuscript that addresses the points raised during the review process.

While the article has undergone major revisions with much success, a few points still require authors' attention. In your revisions, please address reviewer's 2 comments regarding the need to better articulate this article's contribution to the literature on vaccine management systems. In addition, please refer to the comments on the presentation of data (tables 2 and 4) as well as the need for clarifications regarding the use of interviews in the process of data collection and the notes about formatting, and make the appropriate changes.

We look forward to receiving your revised manuscript.

Kind regards,

Anat Rosenthal

Academic Editor

Journal Requirements:

Additional Editor Comments (if provided):

Reviewers' comments:

Reviewer's Responses to Questions

**Comments to the Author**

1. If the authors have adequately addressed your comments raised in a previous round of review and you feel that this manuscript is now acceptable for publication, you may indicate that here to bypass the “Comments to the Author” section, enter your conflict of interest statement in the “Confidential to Editor” section, and submit your "Accept" recommendation.

Reviewer #1: All comments have been addressed

Reviewer #2: (No Response)

2. Does this manuscript meet PLOS Global Public Health’s publication criteria? Is the manuscript technically sound, and do the data support the conclusions? The manuscript must describe methodologically and ethically rigorous research with conclusions that are appropriately drawn based on the data presented.

Reviewer #1: Yes

Reviewer #2: Partly

3. Has the statistical analysis been performed appropriately and rigorously?

Reviewer #1: I don't know

Reviewer #2: No

4. Have the authors made all data underlying the findings in their manuscript fully available (please refer to the Data Availability Statement at the start of the manuscript PDF file)?

Reviewer #1: Yes

Reviewer #2: Yes

5. Is the manuscript presented in an intelligible fashion and written in standard English?

Reviewer #1: Yes

Reviewer #2: No

6. Review Comments to the Author

Reviewer #1: The abstract (content and style) on the cover page is different from the Abstract included in the manuscript. Please reconcile.

Reviewer #2: The writing and tables in this article have been improved, but this paper still needs major work before it is ready for publication. Really articulating what this paper is adding to the existing literature will be very helpful in showing what this important work is adding.

--This paper would benefit from more information in the introduction in terms of what this study adds specifically to the literature on vaccine management systems. Then, this should be clearly articulated in the conclusion.

--If interview data are used in the authors' thinking, that is wonderful. If so, that interview data should be presented in the results, not discussion. They should be integrated in the discussion with quotes, for example in 3.4. If they are not used in the paper, that should be noted in the methods.

--The paper needs major attention to formatting and proofreading. For example, an abstract seems to be have been inserted into the methods on p. 6. A careful proofread of the entire document is needed. There are many grammatical errors and some references are not formatted correctly.

--p. 4, interesting information on the sharp decrease in immunization coverage in 2022. Can this be drawn out a bit more, as the issues listed did not really seem to explain it (wouldn’t these have been issues before)? Is this about COVID, or some political dynamics? How is this related (or not related) to the topic of this paper?

--The formatting of Table 2 still appears incorrect. The percentages seem to be wrong across the columns. How can less than half of all facilities have refrigerators, but then it somehow adds up to 100%?

--Throughout the results, rather than repeating what is in the tables, I suggest using the narrative to hit on and discuss the key points, perhaps with additional inforamtion from the interviews. Much of the discussion could be shortened (eg., it’s fine to just state that refrigerators are everywhere and then move on).

--The formatting of Table 4 is still very confusing; can this be reformatted for clarity?

7. PLOS authors have the option to publish the peer review history of their article (what does this mean?). If published, this will include your full peer review and any attached files.

**Do you want your identity to be public for this peer review?** For information about this choice, including consent withdrawal, please see our Privacy Policy.

Reviewer #1: No

Reviewer #2: No

---

## [Editor Report · Decision Letter 2]

26 Aug 2024

Assessments of Effectiveness of Technologies Utilizations in VIHSCM Among Selected Health Facilities in Tanzania Mainland

PGPH-D-23-02110R2

Dear Professor Mollel,

We are pleased to inform you that your manuscript 'Assessments of Effectiveness of Technologies Utilizations in VIHSCM Among Selected Health Facilities in Tanzania Mainland' has been provisionally accepted for publication in PLOS Global Public Health.

Best regards,

Anat Rosenthal

Academic Editor